# Molecular Weight Identification of Compounds Involved in the Fungal Synthesis of AgNPs: Effect on Antimicrobial and Photocatalytic Activity

**DOI:** 10.3390/antibiotics11050622

**Published:** 2022-05-05

**Authors:** Edward Hermosilla, Marcela Díaz, Joelis Vera, Amedea B. Seabra, Gonzalo Tortella, Javiera Parada, Olga Rubilar

**Affiliations:** 1Chemical Engineering Department, Universidad de La Frontera, Temuco 4811230, Chile; gonzalo.tortella@ufrontera.cl (G.T.); javiera.parada@ufrontera.cl (J.P.); 2Biotechnological Research Center Applied to the Environment (CIBAMA-BIOREN), Universidad de La Frontera, Temuco 4811230, Chile; marcela.diaz@ufrontera.cl (M.D.); j.vera12@ufromail.cl (J.V.); 3Programa de Doctorado en Ciencias de la Ingeniería, Universidad de La Frontera, Temuco 4811230, Chile; 4Center for Natural and Human Sciences, Universidade Federal do ABC, Santo André 09210-580, Brazil; amedea.seabra@ufabc.edu.br

**Keywords:** antimicrobial, photocatalysis, fungal synthesis, synthesis mechanism, silver nanoparticles

## Abstract

The biological synthesis of silver nanoparticles (AgNPs) for medical, environmental, and industrial applications is considered an alternative to chemical synthesis methods. Additionally, the reducing, capping, and stabilizing molecules produced by the organisms can play a key role in the further activity of AgNPs. In this work, we evaluated the synthesis of AgNPs by four molecular weight fractions (S1: <10 kDa, S2: 10 to 30 kDa, S3: 30 to 50 kDa, and S4: >50 kDa) of mycelia-free aqueous extract produced by the white-rot fungus *Stereum hirsutum* and their effect on the antimicrobial activity against *Pseudomonas syringae* and photocatalytic decolorization of nine synthetic dyes exposed to sunlight radiation. All synthesis assay fractions showed the characteristic surface plasmon resonance (SPR) with 403 to 421 nm peaks. TEM analysis of synthesized AgNPs showed different sizes: the whole mycelia-free extracts S0 (13.8 nm), S1 (9.06 nm), S2 (10.47 nm), S3 (22.48 nm), and S4 (16.92 nm) fractions. The results of disk diffusion assays showed an inverse relation between antimicrobial activity and the molecular weight of compounds present in the mycelia-free aqueous extract used to synthesize AgNPs. The AgNPs synthesized by S0 (14.3 mm) and S1(14.2 mm) generated the highest inhibition diameter of *P. syringae* growth. By contrast, in the photocatalytic assays, the AgNPs synthesized by the S2 fraction showed the highest discoloration in all the dyes tested, reaching 100% of the discoloration of basic dyes after 2 h of sunlight exposure. The maximum discoloration observed in reactive and acid dyes was 53.2% and 65.3%, respectively. This differentiation in the antimicrobial and photocatalytic activity of AgNPs could be attributed to the capping effect of the molecules present in the extract fractions. Therefore, the molecular separation of synthesis extract enables the specific activities of the AgNPs to be enhanced.

## 1. Introduction

The production of silver nanoparticles (AgNPs) has emerged due to their wide range of applications in the medical, agriculture, environmental, and industrial fields [1]. The potential of AgNPs is closely related to their nanoscale size, size distribution, shape, particle morphology, particle capping/coating, and optical, electrical, thermal, and electrical conductivity properties [2]. There are several physical and chemical methods (i.e., thermal decomposition, microwave processing, laser ablation, photochemical reduction, chemical reduction, gamma radiation, and evaporation–condensation, among others) that have been developed to synthesize AgNPs; however, they are inefficient, costly, and generate toxic wastes [3,4]. By contrast, the green synthesis of AgNPs by biological reduction of metals is considered a relatively simple, clean, non-toxic, and eco-friendly alternative to chemical methods [5]. The green synthesis of AgNPs has been performed using different organisms, including bacteria, fungi, and plants. These organisms produce metabolites that act as metal-reducing and nanoparticle-capping agents [6,7]. Filamentous fungi have been used in many studies concerning the synthesis of AgNPs due to their high and controlled biomass production and easy handling. The AgNPs synthesis using fungi comprises the preparation of an aqueous extract of fungal biomass, which can reduce Ag^+^ from salt precursors (AgNO_3_ mainly) to elemental silver (Ag^0^) at a nanoscale size (1 to 100 nm). The exact mechanism of AgNP synthesis by fungi involved has not yet been clarified. However, it has been reported that some extracellular metabolites produced by fungi, such as anthraquinones, amino acids, nicotinamide adenine dinucleotide (NADH), polysaccharides, nitrate reductase, extracellular proteins, and enzymes [8,9,10], could participate in metal reduction, capping, and stabilization during the formation of nanoparticles, preventing particle agglomeration and aggregation [6,7]. The Ag^+^ reduction occurs by the combined action of different compounds present in the extract. In this sense, Li et al. (2012) reported that dialyzed extract of *Aspergillus terreus* (7 kDa cut-off) lost its activity to synthesize AgNPs, while it recovered its activity when NADH was added to the dialyzed extract [11]. A prominent hypothesis for the biosynthesis of silver nanoparticles by fungi suggests the combined participation of the enzyme nitrate reductase (a protein of 92 to 100 kDa) and coenzyme nicotinamide adenine dinucleotide phosphate (NADPH) in the process [12,13].

Nevertheless, under certain conditions, NADPH alone can synthesize AgNPs [12]. The SDS-PAGE technique has been used to determine the molecular weight of proteins associated with the extract synthesis and the protein capping of AgNPs. Proteins of 75, 122, 191, and 328 kDa were observed in the extract and covering of AgNPs synthesized using *Aspergillus* sp. [14]. Similar results were shown in SDS-PAGE electrophoresis of AgNPs synthesized using *Macrophomina phaseolina*: an 85 kDa protein was observed in both extract and nanoparticles [10]. The *Trichordema harzianum* extract of synthesis and capping AgNPs showed 36 kDa and 40 kDa proteins [6]. The metabolites that participate as reducing and capping agents in the synthesis process also affect the antimicrobial activity and reactivity of AgNPs [6,15,16]. Guilger-Casagrande et al. (2021) reported that biological AgNPs lost their antimicrobial activity when the nanoparticle capping was removed [15].

Other applications of AgNPs underscore the photocatalytic discoloration of toxic chemical dyes generated in effluent by the textile, plastic, paper, and pharmaceutical industries [17]. Biological synthesized AgNPs under sunlight, UV light radiation, and NaBH_4_ have shown catalytic activity in the degradation of synthetic dyes (methylene blue, methyl orange, rhodamine b, and malachite green, among others) and dye effluents [17,18,19]. Although the effect of capping and stabilizing agents of biological reduction of silver on photocatalytic activity of AgNPs is not clear, the capping agents can enhance the interaction of the dye on the nanoparticle surface [18]. Furthermore, capping agents could positively affect the photocatalytic activity of nanoparticles since they impede the overgrowth and aggregation of nanoparticles [20].

The present work was focused to identify the molecular weight of compounds that participate in the synthesis of AgNPs by the extract of the white-rot fungus, *Stereum hirsutum*, and their effect on antimicrobial activity and photocatalytic discoloration of synthetic dyes. For this purpose, the mycelia-free aqueous extract of *S. hirsutum* was separated into four molecular weight fractions (S1: <10 kDa, S2: 10 to 30 kDa, S3: 30 to 50 kDa, and S4: >50 kDa), which were used to synthesize AgNPs.

## 2. Results and Discussion

In previous work, the white-rot fungus *Stereum hirsutum* was used to synthesize copper and copper-oxide nanoparticles using the extracellular method [3]. In this work, the mycelia-free aqueous extract of *S. hirsutum* (S0) showed effective and efficient activity for synthesizing AgNPs using AgNO_3_ as a salt precursor. Fungi secrete metabolites (NADH, NADPH, peptides, naphthoquinones, anthraquinones, proteins, and enzymes) during the preparation of the extract, which can play a key role in the AgNPs synthesis. However, the exact mechanism of AgNPs synthesis by the fungi involved has not been fully explained yet [6,12]. This work was focused to identify the molecular weight of the metabolites responsible for the synthesis of AgNPs by *S. hirsutum* through a molecular weight fractionation of the mycelia-free aqueous extract.

### 2.1. SDS-PAGE Electrophoresis of the Mycelia-Free Extract of Stereum hirtusum

The SDS-page electrophoresis of the mycelia-free extract revealed that it is composed of a set of proteins of different molecular weights (from 29 to 181 kDa), with most marked bands at 29 kDa, 42 kDa, 49 kDa, and 59 kDa (Figure 1a). Rodrigues et al. (2013) showed that 75, 122, 191, and 328 kDa proteins were observed in the extract of *Aspergillus* sp. used to synthesize AgNPs [14]. By contrast, the 85 kDa protein was detected in the extract of *Macrophomina phaseolina* [10]. The *Trichoderma harzianum* extract of synthesis showed 36 kDa and 40 kDa proteins [6]. These reports suggest that these proteins act as reducing or capping agents in the synthesis of the AgNPs. Therefore, to identify the molecular weight of compounds involved in the synthesis, the S0 extract of *S. hirsutum* was sequentially filtered through Amicon tubes with cut-offs of 10 kDa, 30 kDa, and 50 kDa, as shown in Figure 1b. Four molecular weight fractions were obtained from S1 with compounds < 10 kDa, S2 with compounds between 10 and 30 kDa, S3 with compounds between 30 and 50 kDa, and S4 with compounds between 30 and 50 kDa. These molecular fractions were used to synthesize AgNPs using AgNO_3_ as a salt precursor.

### 2.2. X-ray Diffraction (XRD) Analysis of the Synthesized AgNPs

XRD analysis was used to corroborate the formation of the AgNPs by the mycelia-free extract of *S. hirsutum* (Figure 2). The diffraction peaks observed at 38.11°, 44.27°, 64.42°, and 77.48°, corresponding to (111), (200), (220), and (311) of the face-centered cubic structure of metallic silver, matched with the reference of the FCC structure of the Joint Committee of Powder Diffraction Standard (JCPDS file no. 89-3722) from PDF-2 database. The XRD spectra also exhibited the presence of an additional peak before 30°, which might be due to the presence of organic compounds in the sample [21].

### 2.3. UV-Vis Spectroscopy of AgNPs

All the extract fractions changed the reaction mixture from colorless to a dark brown solution with different color intensities, which indicates a successful synthesis of AgNPs. However, the change of color observed in the reaction mixture with a whole mycelia-free aqueous extract (S0) and the S1 fraction was faster (a couple of minutes) than the S2, S3, and S4 fractions (between 2 and 3 h), which indicates that smaller compounds (<10 kDa) are more efficient as Ag^+^ reducing agents than larger compounds (>10 kDa). The resultant solutions after 24 h are shown in Figure 3a. Li et al. (2012) showed that <7kDa compounds of *A. terreus* extract play a key role in the synthesis of AgNPs [11]. UV-vis spectroscopy is widely used to corroborate the formation of surface plasmon resonance of AgNPs between 400 and 450 nm [22]. Figure 3b shows the UV-vis spectra obtained from AgNP solutions synthesized by the different extract fractions. The SRP peak of synthesized AgNPs varied from 403 to 421 nm. Similar SRP peaks were observed in the S0, S1, and S3 extract fractions at 421 nm, 417 nm, and 413 nm, respectively (Table 1). A significant decrease in the SRP peak to 403 nm for both the S2 and S4 fractions with a higher absorbance intensity and sharper SRP than the other fractions was observed, suggesting a higher synthesis yield and the presence of spherical or round nanoparticles in the dispersions [17].

Guilger-Casagrande et al. (2021) observed that the capping of AgNPs decreases the SRP peak by a couple of nanometers compared with uncapped AgNPs; therefore, the difference in SRP peaks observed in synthesized AgNPs can be attributed to different capping compounds [6]. Data of UV-vis spectra were used to calculate the bandgap of AgNPs by the Tauc plot [23]. The AgNPs synthesized by the S0, S1, and S4 fractions showed similar band gap values of 2.61 eV and 2.62 eV, respectively. A slight increase in bandgap values to 2.76 eV, 2.76 eV, and 2.61 eV was observed in AgNPs synthesized by S2, S3, and S4 extracts. The AgNPs have shown an optical bandgap ranging from 2.4 to 3.4 eV [24].

### 2.4. Particle Size and Zeta Potential

The AgNPs synthesized by molecular fractions showed different hydrodynamic sizes (Table 1). S0 and S1 showed larger sizes of 79.3 nm and 66.00 nm, respectively. S2, S3, and S4 showed similar sizes of 19.1 nm, 17.0 nm, and 14.3 nm, respectively. Therefore, as the molecular weight of the compounds of mycelia-free extract increases, the hydrodynamic size of synthesized AgNPs decreases. However, the sizes of the AgNPs determined by TEM for the S0 (13.8 nm) and S1 (9.06 nm) fractions differ significantly from the size results measured by DLS (Table 1). By contrast, a similar size was observed in AgNPs synthesized by the S2 (10.47 nm), S3 (22.48 nm), and S4 (16.92 nm) fractions compared with DLS sizes (Table 1 and Figure 3e–g). The differences observed between TEM and DLS diameters in AgNPs synthesized by the S0 and S1 fractions could indicate nanoparticle aggregation in the water suspensions due to capping compounds in the hydrodynamic sphere [25]. According to the statistics test, the AgNPs synthesized by the S1 and S2 fractions showed the smallest nanoparticle size, whereas the S3 fractions showed the largest size. The TEM images reveal that the nanoparticles synthesized by both S0 and fractions have a spherical or quasi-spherical shape (Figure 3c–g). The zeta potential of synthesized AgNPs ranged from −33.1 mV to −44.8 mV. The zeta potential of AgNPs synthesized by S0 was significantly lower than AgNPs synthesized by the molecular fractions. No significant differences were observed in the zeta potential of nanoparticles synthesized by S1, S2, S3, or S4 fractions (Table 1). The zeta potential can be influenced by the capping agents on the surface of nanoparticles.

The zeta potential value indicates the surface charge potential, which is directly related to the stability of nanoparticles in suspension. In the literature, it has been stated that zeta potential values higher than +30 mV or lower than −30 mV indicate that nanoparticles are very stable in the dispersion solvent [26]. Thus, AgNPs synthesized by the molecular fractions could be more stable than AgNPs synthesized by S0.

### 2.5. Antimicrobial Activity

The antimicrobial activity of the AgNPs synthesized by molecular fractions was evaluated against the Gram-negative bacterium *Pseudomonas syringae* using the disk diffusion method. *P. syringae* is a cosmopolitan phytopathogen that causes bacterial canker disease and can attack the stones of growing fruit, generating considerable damage to trees and reducing yield and economic losses [27]. The results of disk diffusion assays showed an inverse relationship between antimicrobial activity and the molecular weight of compounds present in the mycelia-free aqueous extract used to synthesize AgNPs (Figure 4).

The AgNPs synthesized by S0 (14.3 mm) and S1(14.2 mm) generated the highest inhibition diameter of *P. syringae* growth. S2 slightly decreased the inhibition diameter to 12.6 mm with no statistical differences with S0 and S1. The AgNPs synthesized by S3 and S4 fractions showed lower inhibition diameters of 10.4 mm and 8.7 mm, respectively. Similar inhibition diameters of *P. syringae* were obtained with AgNPs synthesized by the plant extract of *Prunus cerasifera* [28]. Although the size of AgNPs synthesized by fractions S2, S3, and S4 are significantly smaller than AgNPs synthesized by S0 and S1, the nanoparticle size does not appear to have an effect on its antimicrobial activity. AgNPs synthesized by the extract of the white-rot fungus *Ganoderma applanatum* of 133 nm showed high inhibition zones (up to 28 mm) against human-pathogen bacteria [29]. These results indicate that when compounds of molecular weight >10 kDa participate in the synthesis of AgNPs, the antimicrobial activity decreases significantly. It has been reported that capping affects the antimicrobial activity of AgNPs [6]. Therefore, the capping agents in the S0 and S1 fractions have higher antimicrobial activity than those in the S2, S3, and S4 fractions. As noted, the formation of AgNPs by fractions S0 and S1 was faster than with other fractions, indicating that low molecular weight compounds (<10 kDa) can better reduce the activity of silver ions. These compounds act as capping agents faster than higher molecular weight compounds present in the whole mycelia-free aqueous extract (S0). The minimum inhibitory concentration against *P. syringae* of AgNPs synthesized by the mycelia-free extract of *Stereum hirsutum* (S0) was 40 μg mL^−1^ (unpublished data).

To understand the antimicrobial activity of the AgNPs, three modes of action have been proposed: (1) cell wall and membrane damage, (2) intracellular penetration and damage, and (3) oxidative stress [13]. The cell wall of Gram-negative bacteria such as *P. syringae* contains at least two lipopolysaccharide layers that protect the bacteria against environmental threats, maintain cell homeostasis, and transport nutrients inside the cell. The AgNPs release Ag^+^ ions which damage the cell wall and membrane, increasing their permeability; as a result, the cellular contents (proteins, enzymes, DNA, ions, and other metabolites) are released into the environment [13,30]. In addition, the AgNPs can also penetrate the periplasmic space and ultimately be inside the cell, where they can cause inhibition of DNA replication, inactivation of proteins and enzymes, disassembly of ribosome units, and oxidative stress, affecting crucial and vital cell functions [31]. Figure 5 shows a schematic representation of the main mechanism involved in the antimicrobial activity of AgNPs.

### 2.6. Photocatalytic Discoloration of Synthetic Dyes

Synthetic dyes are among the most produced water pollutants by the textile, plastic, paper, food, tanneries, and pharmaceutical industries. These dyes are highly resistant to degradation due to their chemical structure. In aquatic systems, synthetic dyes are toxic and decrease sunlight’s penetration, affecting aquatic organisms [17]. Here, we evaluated the catalytic activity of the synthesized AgNPs for the discoloration of nine water-soluble synthetic dyes under sunlight radiation for two hours. Three dyes are classified as acid dyes, four as basic dyes, and two as reactive dyes. Figure 6 shows the percentage of dye discoloration catalyzed by AgNPs. The AgNPs synthesized using the S2 fraction showed the highest discoloration percentages on all the dyes tested, followed by AgNPs synthesized by the S3 fraction. Acid blue 1, acid orange 6, and acid red 27 reached a maximum discoloration of 53.2%, 25.3%, and 50.7% by AgNPs synthesized by the S2 fraction. Acid orange 6 was only discolored by AgNPs synthesized by the S2 fraction. AgNPs synthesized by the S4 fraction showed the lowest degradation rates of acid dyes. Concerning basic dyes, basic blue 24, basic blue 41, and basic violet 4 showed complete discoloration catalyzed with AgNPs synthesized by the S2 fraction, while basic blue 41 was also completely discolored with AgNPs synthesized by the S3 fraction. The maximum discoloration of basic orange 2 (53.7%) was also obtained with AgNPs synthesized by the S2 fraction. The reactive dyes, reactive blue 5 and reactive blue 19, showed 54.9% and 65.1% of discoloration by AgNPs synthesized by the S2 fraction, respectively. Interestingly, we note that compounds from mycelia-free aqueous extract of *S. hirsutum* with a molecular weight ranging from 10 kDa to 30 kDa (S2 fraction) significantly increased the photocatalytic activity of AgNPs for discoloring synthetic dyes, compared with AgNPs synthesized with the whole mycelia-free aqueous extract. Similar results were observed in the photocatalytic activity of AgNPs synthesized with compounds with molecular weights in the range of 30 kDa to 50 kDa (S3 fraction). By contrast, the synthesis of AgNPs with compounds < 10 kDa (S1 fraction) and >50 kDa (S4 fraction) showed equivalent or worse discoloration rates of synthetic dyes than AgNPs synthesized with the whole mycelia-free aqueous extract. These results suggest that the photocatalytic activity of biologically synthesized AgNPs could be affected by the reducing and capping agents present in the extract or its fractions. Sumi et al. (2017) reported that the dye forms a complex with the capping agent at the nanoparticle surface, which favors the exchange of electrons between the dye and the nanoparticle and, therefore, its degradation [18]. It has also been reported that the size, structure, crystallographic nature, and morphology of AgNPs affect the photocatalytic degradation of synthetic dyes [32]. The performance of a photocatalyst also depends on its electronic band structure and bandgap energy. An efficient photocatalyst should have a bandgap energy ranging from 2 to 3 eV to absorb light in the visible region (400 to 700 nm) provided by sunlight [33]. The bandgap of synthesized AgNPs by mycelia-free aqueous extract fractions was 2.61 eV to 2.76 eV. Therefore, all synthetized AgNPs can act as photocatalysts under sunlight radiation. AgNPs can absorb the full solar light spectrum during photocatalysis, including UV (200–400 nm) and visible (400–700 nm) light ranges. When AgNPs absorb the visible spectrum of sunlight, the electrons from the outer sp band are excited to a higher energy state due to the SPR effect [18]. The SPR effect is the collective oscillation of electrons in the conduction band in the nanoparticles, which resonates with the electromagnetic field of the incident light. O_2_ molecules accept these excited electrons to form superoxide radicals (O_2_•). The produced radicals attack and break the dye molecules adsorbed on the nanoparticle surface. The process of dye adsorption on the nanoparticle surface can be favored by the capping agents. In addition, the electron holes (h+) generated in the 5sp orbital of Ag atoms accept electrons of the dye molecule, degrading the dye. By contrast, UV light is absorbed by AgNPs due to the interband transition of electrons from the 4d orbital to the 5sp [34]. This transition between bands leads to the excitation of many photogenerated electrons. These excited electrons interact with oxygen molecules to form oxygen radicals (O_2_•) and the hydroxyl ion to form hydroxyl (•OH) radicals, attacking and degrading the dye molecule. Again, the holes generated in the 4d orbital of the AgNPs accept electrons of the dye molecule, leading to further degradation. The photocatalysis mechanism of AgNPs under UV and visible light radiation for the dye degradation are in Figure 7.

In relation to the industrial application of the synthetic dyes used in this work, acid dyes are applied to protein fibers, nylon, wool, or silk, reactive dyes mainly used for dyeing cotton fabrics. In contrast, basic dyes are usually applied to acrylic, paper, and nylon substrates [35]. The basic dyes produce colored cations in the solution, while reactive and acid dyes produce anions. This feature of basic dyes makes them more easily attracted to the negatively charged surface of AgNPs for further degradation, which could explain the higher discoloration of basic dyes than the acid and reactive ones observed in the photocatalytic assays. Therefore, the synthesized AgNPs by the S2 fraction of the *S. hirsutum* extract have an attractive potential to be applied to the photocatalytic treatment of dyes or effluents from the textile industry, where basic dyes are mainly used. However, it is essential to evaluate the potential effects of radicals on the capping agents of AgNPs and alternatives of AgNP reusability in further studies.

### 2.7. Molecular Weight of the Compounds Involved in the AgNPs Synthesis

The white-rot fungi such as *S. hirsutum* can secrete large amounts of metabolites and proteins which play a crucial role in their life cycle [36]. This features difficult-to-establish specific mechanisms and metabolites involved in the synthesis of AgNPs by this type of organism. Some low molecular weight metabolites can include NADH, NADPH, phenolic compounds, organic acids, among others. The secreted proteins by white-rot fungi are mainly hydrolases (amylases, cellulases, proteases, and lipases) and oxidoreductases (laccases and peroxidases). In this study, the mycelia-free extract of *S. hirsutum* and its molecular weight fractions showed that compounds with different molecular weights have the potential to synthesize AgNPs under alkaline conditions (pH 12) using AgNO_3_ as the salt precursor. Figure 8 summarizes our findings. A first separation of the compounds < 10 kDa (non-protein compounds) and >10 kDa (proteins) allows identification of the nature of the metabolites that can participate in the Ag^+^ reduction, stabilization, and nanoparticle capping during the synthesis process. We observed that the process of AgNPs synthesis by compounds < 10 kDa is faster (a couple of minutes) than the synthesis by compounds with molecular weight > 10 kDa (2 to 3 h). To identify the molecular weights of the proteins that have the potential to synthesize AgNPs, the fungal extract was separated into three protein fractions: S2 (10 to 30 kDa), S3 (30 to 50 kDa), and S4 (>50 kDa), which were used for AgNPs synthesis. SDS-PAGE electrophoresis showed that the fungal extract has proteins of 29 to 181 kDa, with intense bands of 29, 42, 49, and 59 kDa. Therefore, the AgNPs synthesis by S2 fraction could be carried out by a 29 kDa protein. The formation of AgNPs by the S3 fraction could result from the participation of two main proteins (42 and 49 kDa), and other proteins (32, 35, and 37 kDa). A 59 kDa protein and other proteins (69, 73, 82, 102, and 181 kDa) in the S4 fraction could participate in the AgNPs synthesis. A study showed that in the synthesis of AgNPs by *Aspergillus flavus,* a 32 kDa protein, participates in the Ag^+^ reduction and a 35 kDa acts as a capping agent of nanoparticles [36]. Similarly, two proteins (36 and 40 kDa) were detected in the extract and AgNPs synthesized by *Trichoderma harzianum*, which were identified as b-1,3-glucanase and chitinase enzymes, respectively [6]. We evaluated the antimicrobial activity against *P. syringae* and photocatalytic discoloration of synthetic dyes by the synthesized AgNPs with the molecular weight fractions. The results showed that the AgNPs synthesized by <10 kDa have high antimicrobial activity but low photocatalytic activity. On the other hand, AgNPs synthesized by S2, and S3 fractions showed high photocatalytic activity for the degradation of synthetic dyes (mainly basic dyes) but low antimicrobial activity. Interestingly, the AgNPs synthesized by the S4 fraction showed low antimicrobial and photocatalytic activities. Hence, the fungal compounds < 10 kDa confer antimicrobial activity to AgNPs. It has been reported that capping agents such as aromatic compounds and organic acids can enhance the antimicrobial activity of AgNPs synthesized by green methods. Compounds of 10 to 50 kDa could promote the photocatalytic activity of synthesized AgNPs due to nanoparticle surface modification. The AgNPs synthesized by the S2 fraction showed the lowest zeta potential (−44.8 mV), which favors the interaction of the dye on the nanoparticle surface and prevents the agglomeration of AgNPs.

## 3. Materials and Methods

### 3.1. Culture of Microorganisms

The white-rot fungus *S. hirsutum* was obtained from the culture collection of the environmental biotechnology laboratory at the Universidad de La Frontera, Chile. This fungal strain (code CCCT22.02) was isolated from decaying oak wood [3] and deposited in the Chilean Culture Collection of Type Strains, Scientific and Technological Bioresource Nucleus at Universidad de La Frontera (Chile). The fungus was maintained in plates containing potato dextrose agar (PDA) at 4 °C and periodically sub-cultured. The Gram-negative bacterium *Pseudomonas syringae* (code: CCCT22.03) was isolated from a peach stalk tree and deposited in the Chilean Culture Collection of Type Strains, Scientific and Technological Bioresource Nucleus at Universidad de La Frontera (Chile). *P. syringae* was maintained in Mueller Hinton Broth at 4 °C and periodically sub-cultured.

### 3.2. Preparation and Fractionation of the Mycelia-Free Aqueous Extract

Five agar plugs of *S. histutum* were transferred to 250 mL Erlenmeyer flasks with 100 mL of liquid medium (containing: 15 g L^−1^ glucose, 5 g L^−1^ peptone from potatoes, and 2.5 g L^−1^ yeast extract), which were incubated in the dark under static conditions for two weeks at 25 °C. After that, the fungal biomass was separated from the culture by vacuum filtration through Whatman No 1 filter paper and washed with deionized water. The mycelia-free aqueous extract was prepared by transferring the fungal biomass (628 mg dry weight) into 100 mL Erlenmeyer flasks containing 50 mL of deionized water and incubated in an orbital shaker at 100 rpm for 24 h at 25 °C. Finally, the supernatant (mycelia-free aqueous extract) was recovered by vacuum filtration, and the fungal biomass was discarded. Figure 1b schematizes the molecular weight fractionation of the mycelia-free aqueous extract of *S. hirsutum*. The 15 mL of the prepared mycelia-free aqueous extract was firstly filtered in a 10 kDa Amicon tube, the S1 fraction was recovered from the filtrate, and the concentrate fraction was placed in a 30 kDa Amicon tube and filtered. Again, the filtrate was recovered (S2 fraction), and concentrate was placed into a 50 kDa Amicon tube; the filtrate (S3 fraction) and concentrate (S4 fraction) were recovered. For the filtering processes, the Amicon tubes were centrifugated at 4100 rpm for 30 min to obtain a filtrate (compounds with molecular weight lesser than cut-off) and concentrate (compounds with molecular weight higher than cut-off). The volume of each fraction was adjusted to 15 mL before the synthesis of AgNPs by adding deionized water. The mycelia-free aqueous extract without molecular fractionation (S0) was used as a control for synthesis. All trials were carried out in triplicate.

### 3.3. Synthesis of AgNPs

The AgNPs were synthesized in a reaction mixture containing 15 mL of the extract fraction and AgNO_3_ (3 mM) as a precursor salt. The reaction was kept under magnetic agitation at room temperature. The pH was adjusted to 12 by adding drops of NaOH (4 M). The nanoparticle dispersions were shaken at 150 RPM for 7 days before characterization. The obtained particles were frozen to −80 °C and then freeze-dried. The dried AgNPs powder was stored for further assays.

### 3.4. Characterization of AgNPs

The absorption spectra of the AgNPs in the range of 200 to 800 nm were obtained using a spectrophotometer (Genesis^TM^ 10 S UV-Vis, Thermo Scientific, Waltham, MA, USA). The optical gap band was estimated through a Tauc plot according to Mistry et al. (2021). Dynamic light scattering analysis (DLS) was carried out using a Zetasizer Nano ZS90 System (Malvern Instruments, Malvern Instruments, Malvern, UK) to measure the hydrodynamic diameter and zeta potential of the synthesized AgNPs. Transmission electronic microscopy (TEM) analysis of synthesized AgNPs was performed in a Carl Zeiss LIBRA^®^ 120 PLUS microscope. For the TEM analysis, drops of AgNPs solutions were placed in a carbon-coated copper grid and observed at 120 kV. The size distribution of AgNPs was measured from TEM images employing Sigma Scan Pro 5 software.

### 3.5. Antimicrobial Activity of AgNPs

The antimicrobial activity of AgNPs synthesized by the fractions (S0, S1, S2, S3, and S4) of the extract was evaluated against the Gram-negative bacterium *Pseudomonas syringae* using paper disk diffusion. Briefly, sterile Petri dishes with 15 mL of Mueller Hinton medium were spread with 150 μL of a 24-h-old culture of *P. syringae* (1 × 10^5^ CFU mL^−1^) on Mueller Hinton Broth. The paper disks (6 mm diameter) were placed on the plates and loaded with 60 μL of each AgNP solution (50 µg mL^−1^). The plates were incubated at 37 °C, and the antimicrobial activity of AgNPs was assessed by measuring the inhibition diameter around the disks after 24 h of incubation using the Sigma Scan Pro 5 software.

### 3.6. Photocatalytic Dye Discoloration Using AgNPs

Each dye solution’s UV-vis spectra (range 400 nm to 700 nm) were measured before the discoloration assays to obtain the maximum absorption wavelength (λmax). The discoloration assays were performed in 96-well ELISA microplates with three acid dyes, four basic dyes, and two reactive dyes at concentrations shown in Table 2, the absorbance of which at λmax was close to 1.0 and 500 mg L^−1^ of Ag nanoparticles. The microplates were exposed to direct sunlight for two hours. The absorbance value at λmax for each dye was measured at zero and two hours. The percentage of dye discoloration was calculated using Equation (1):(1)$$\mathrm{Dye}\;\mathrm{discoloration}\left(\%\right)=\frac{{\mathrm{Abs}}_{0}-{\mathrm{Abs}}_{1}}{{\mathrm{Abs}}_{0}}\times 100$$
where Abs_0_ corresponds to zero-time absorbance of dye, and Abs_1_ corresponds to the dye absorbance after 2 h of sunlight exposure. The absorbance values were corrected by subtracting the absorbance of dye control (without nanoparticles) exposed to sunlight for 2 h and the absorbance of alone nanoparticles as the control.

The dye concentration was adjusted to an absorbance close to 1.0 at λmax.

### 3.7. The SDS-PAGE Electrophoresis of Mycelia-Free Extract

The characterization of proteins presented in the mycelia-free extract of *S. hirsutum* was carried out by sodium dodecyl sulfate-polyacrylamide gel (SDS-PAGE) electrophoresis [37]. The samples were prepared with 20 uL of extract mixed with 5 uL of SDS-PAGE loading buffer (0.06 M Tris pH 6.8, 10% glycerol, 2% SDS, 0.1% bromophenol blue, and 1.5% DDT), then heated at 100 °C for 5 min. The sample and protein ladder of broad molecular mass range RGB Plus was prestained (245 kDa to 8 kDa), and Maestrogen were separated in a 10% SDS-polyacrylamide gel using a vertical electrophoresis CZ24 apparatus (Hangzhou LongGene Scientific Instrument Co., Hangzhou, China) at a constant voltage of 200 kV for 40 min. After electrophoresis, the protein bands were stained with Coomassie brilliant blue R-280 for 12 h and destained in a solution of water, methanol, and acetic acid (5:5:1) for 2 h. Afterward, the gel was photographed and analyzed in the freeware GelAnalyzer 19.1 (www.gelanalyzer.com, accessed on 19 February 2022) by Istvan Lazar Jr., Ph.D., and Istvan Lazar Sr., Ph.D., CSc.

### 3.8. XRD Analysis of AgNPs

The X-ray diffraction (XRD) analysis of AgNPs was performed using a STADI-P (Stoe^®^, Darmstadt, Germany) diffractometer operating at room temperature, at 50 kV and 40 mA, and while using MoKα1 (λ = 0.7093 Å) radiation. The X-ray photons were collected by a Mythen 1 K (Dectris^®^, Baden, Switzerland) detector. Data were recorded in a powdered sample loaded into a 0.3 mm diameter special glass 14 capillary (Hilgenberg, Malsfeld, Germany), in the 2θ range from 10° to 80°. Qualitative phase analysis of PXRD data was conducted utilizing the search-match tool of X’pert Highscore software with a version of the PDF-2 database.

## 4. Conclusions

We demonstrated the great potential of *S. hirsutum* to synthesize AgNPs via an extracellular method. It was also demonstrated that compounds of different molecular weights released by the fungus have an efficient activity to reduce Ag ions and stabilize the obtained AgNPs. The AgNPs synthesized by the different molecular fractions of the mycelia-free extract of *S. hirsutum* exhibited different antimicrobial and photocatalytic potential. Conversely, the AgNPs synthesized by <10 kDa compounds presented high antimicrobial activity, while the AgNPs synthesized by compounds between 10 to 30 kDa showed high photocatalytic activity. This differentiation in AgNP activity could be attributed to the capping effect of the molecules present in the fraction. Therefore, the molecular separation of the synthesis extract makes it possible to enhance specific activities of the AgNPs. Based on these results, we conclude that the AgNPs synthesized by *S. hirsutum* can be used in many applications.

## Figures and Tables

**Figure 1 antibiotics-11-00622-f001:**
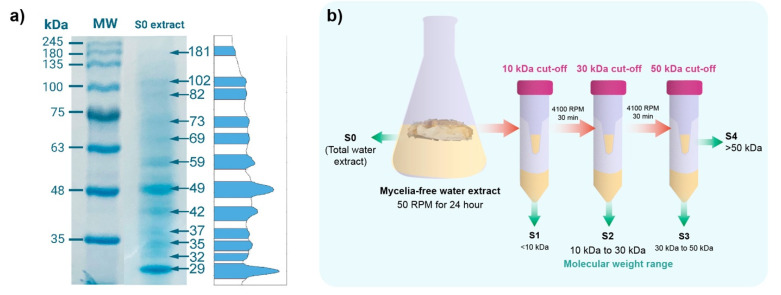
(**a**) SDS-page electrophoresis of the mycelia-free extract (S0) and (**b**) scheme of molecular weight fractionation of mycelia-free aqueous extract to synthesize AgNPs using Amicon ultra tubes. S0 to S4 indicate fractions used for the synthesis of AgNPs.

**Figure 2 antibiotics-11-00622-f002:**
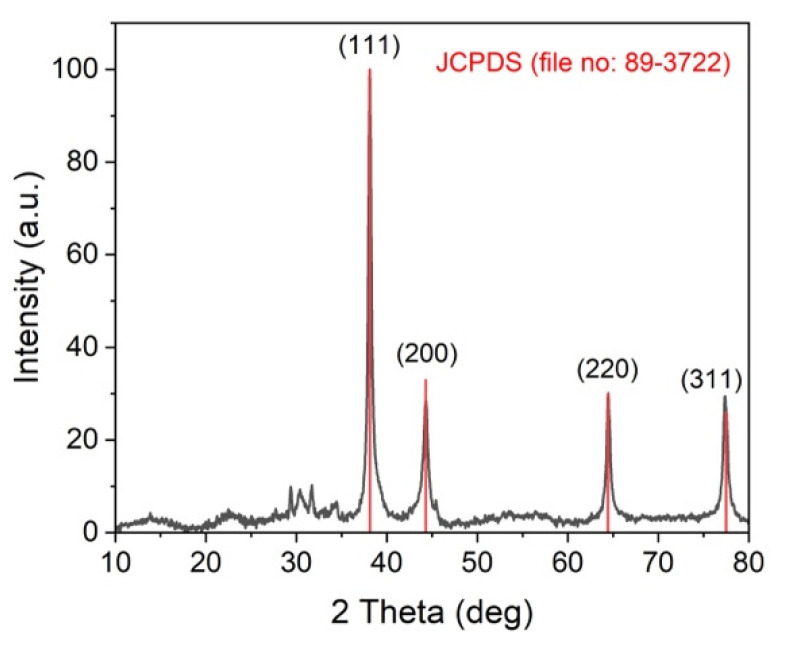
XRD spectra of AgNPs synthesized by the mycelia-free aqueous extract of *S. hirsutum*.

**Figure 3 antibiotics-11-00622-f003:**
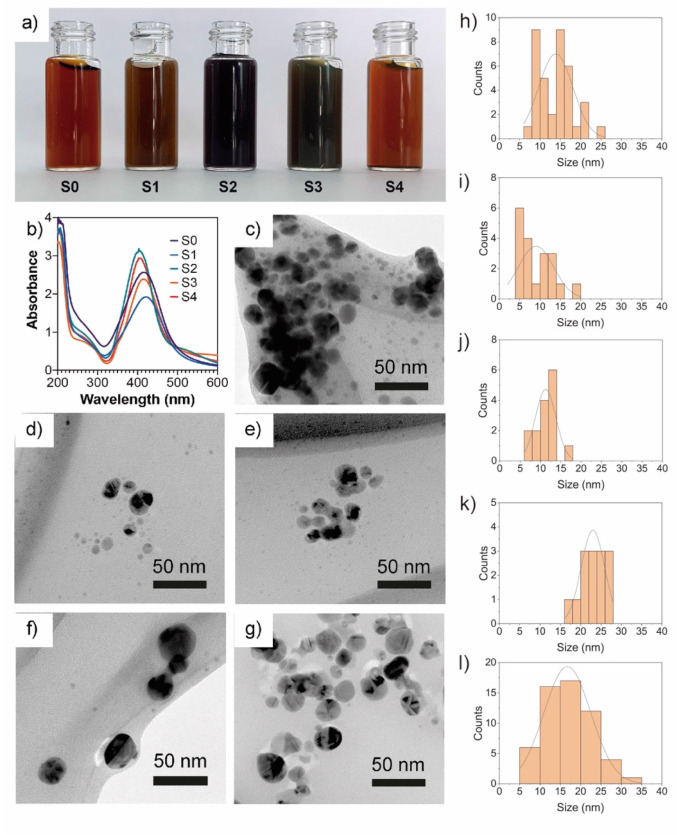
Photography (**a**) of AgNPs dispersions, UV-vis spectra (**b**) of AgNPs obtained using different mycelia-free aqueous extract fractions (all nanoparticle solutions were diluted in water 1:20), TEM images (**c**–**g**), and histograms (**h**–**l**) of AgNPs synthesized with S0, S1, S2, S3, and S4 extract fractions, respectively.

**Figure 4 antibiotics-11-00622-f004:**
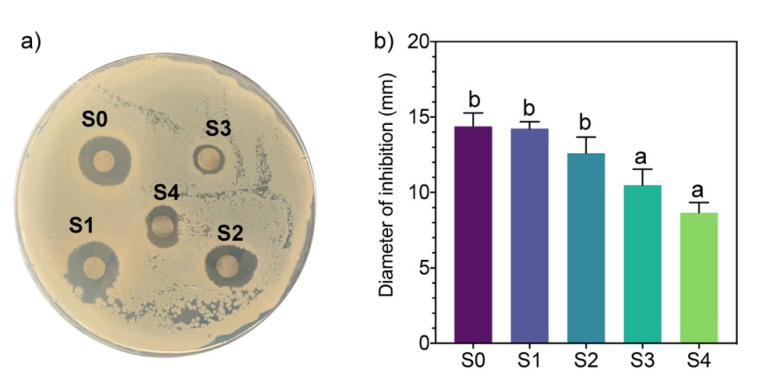
(**a**) Antimicrobial activity of AgNPs synthesized using different mycelia-free aqueous extract fractions by the method of paper disk diffusion. (**b**) Different letters indicate that the values are statistically different.

**Figure 5 antibiotics-11-00622-f005:**
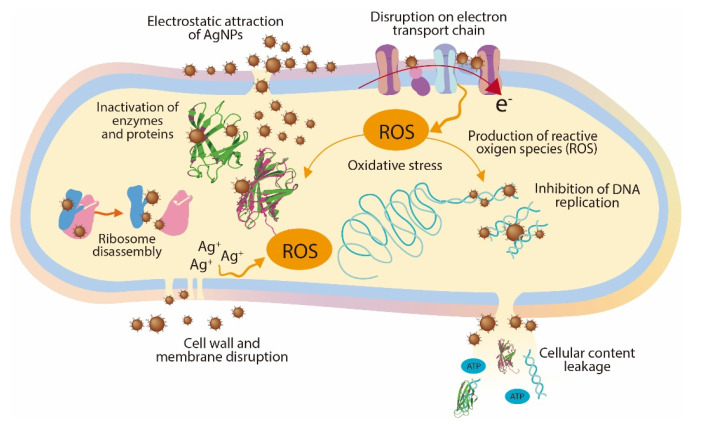
Antimicrobial mechanisms of AgNPs.

**Figure 6 antibiotics-11-00622-f006:**
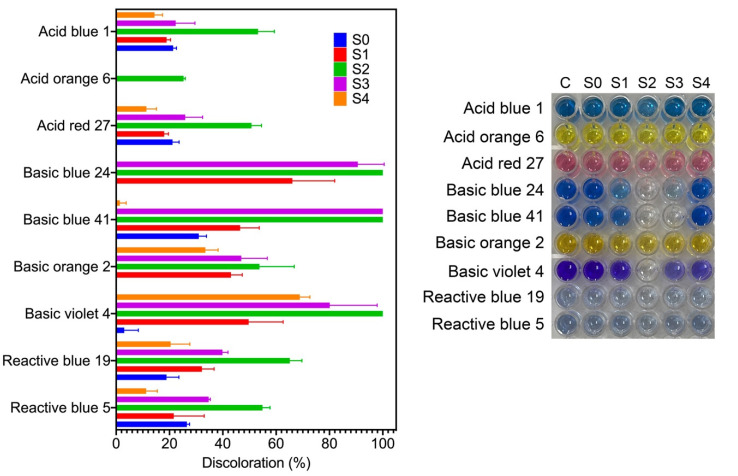
Discoloration of synthetic dyes through photocatalytic reaction using AgNPs obtained using different mycelia-free aqueous extract fractions after 2 h of sunlight exposure.

**Figure 7 antibiotics-11-00622-f007:**
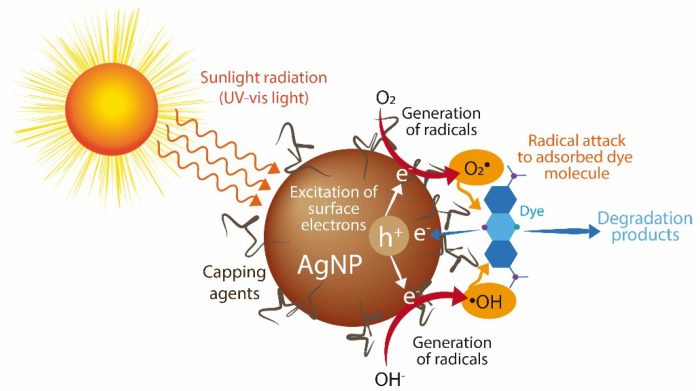
Photocatalysis mechanism of AgNPs under UV and visible light radiation for dye degradation.

**Figure 8 antibiotics-11-00622-f008:**
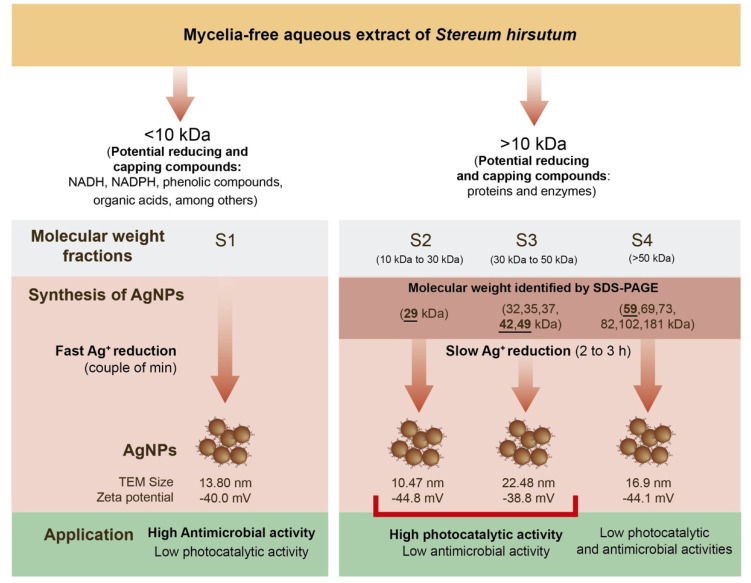
Synthesis of AgNPs by molecular weight fractions from mycelia-free aqueous extract of *S. hirsutum*.

**Table 1 antibiotics-11-00622-t001:** Characterization of the AgNPs obtained using different mycelia-free aqueous extract fractions.

Parameter	Synthesis Fraction				
	S0	S1	S2	S3	S4
SPR peak (nm)	417	421	403	414	403
Particle size DLS (nm)	79.3 ± 30.6b	66.0 ± 10.2b	19.1 ± 4.1a	17.0 ± 5.0a	14.3 ± 2.0a
Zeta potential (mV)	−33.1 ± 2.4b	−40.0 ± 10.4ab	−44.8 ± 8.4a	−38.8 ± 4.2ab	−44.1 ± 2.9a
TEM size (nm)	13.80 ± 4.13c	9.06 ± 4.16a	10.47 ± 3.53ab	22.48 ± 3.46d	16.92 ± 5.77c
Direct gap band (eV)	2.61	2.62	2.76	2.76	2.67

SPR = surface plasmon resonance; DLS = dynamic light scattering; Direct and indirect gap bands were calculated using the Tauc plot method (Appendix A). Different letters indicate that the values are statistically different for the same row.

**Table 2 antibiotics-11-00622-t002:** Concentration and λmax of synthetic dyes used in photocatalytic assays using AgNPs obtained using different mycelia-free aqueous extract fractions.

Synthetic Dye	λmax (nm)	Concentration (μM)
Acid blue 1	610	20
Acid orange 6	590	20
Acid red 27	525	40
Basic blue 24	590	20
Basic blue 3	650	20
Basic blue 41	610	20
Basic orange 2	450	25
Basic violet 4	596	20
Reactive blue 19	600	50
Reactive blue 5	600	32

## Data Availability

Not applicable.

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
