# Peer review of "Molecular Weight Identification of Compounds Involved in the Fungal Synthesis of AgNPs: Effect on Antimicrobial and Photocatalytic Activity"

_antibiotics, 2022, doi:10.3390/antibiotics11050622_

Round 1
Reviewer 1 Report
This manuscript is well deign and performed. It can be published prior to some moidifications
- Introduction will be more clear and reflected citing some new recent references.
- The molecular weight of proteins which are studied in this manuscript by the author should more elaborate of their complete mechanism in the synthesis process and which path they follow.
- English language should be more polished.
- Author should present why choose fungal resources for synthesis of AgNPs.
- Discussion part should be more enriched by citing some suggested references related fungal mediated synthesis of AgNPs such as doi:10.3390/molecules23030655 ,doi:10.1049/iet-nbt.2015.0059,doi: 10.1007/s00449-014-1205-6.
- Author should try to give a graphical abstract.
Reviewer 2 Report
1.Improve the introduction section
2.what is the role of NaOH in nanoparticle synthesis
3..add culture accession number S. histutum
4. how to set the concentration of AgNPs in antimicrobial activity
5.Gram-negative bacterium Pseudomonas syringae obtained from where?? add accession number of Gram-negative bacterium Pseudomonas syringae
6.why particularly choose antibacterial activity of Gram-negative bacterium Pseudomonas syringae
7.what is the positive control used for antibacterial activity
8.improve the discussion
9.what is the average size of the AgNPs size ?? author how to calculate the NPs size?? author should be add the histogram analysis of NPs
10.what is the crystalinity and purity of the AgNPs,author should be do the analysis of XRD and EDX for AgNPs
11.what is the synthesis mechanism of AgNPs,author should be add this content with schematic diagram
12.author should be add the followed statistical analysis
13.what is the reusability of the AgNPs in dye degradation?? author should be do the reusablity analysis and add the results
13.add the dye degradation images
Reviewer 3 Report
Hermosilla et al. study the impact of fungal extracts fractionated by molecular weight on the synthesis of Ag NPs and their activity.
L 13: I have to object. The widespread usage of Ag-NPs is highly controversial as they may infiltrate nearly any tissue and accumulate.
L117: the description of the filtering method is not clear to me. How much filtrate, how is left in the supernatant? Overconcentration and lack of salts in the supernatant may lead to aggregation and thus lower solubility and lower activity, respectively
L 119, 134, 213: Mind the concentrations. What was the total biomass in the extracts S1 - S4? Was the concentration adjusted?
L 158: no description of the TEM sample preparation and imaging method. How were the NPs separated from the organic material? It seems like the NPs are enclosed in organic droplets?
Figure 2 and Table 1: The particles visible on the TEM images and the corresponding sizes given in table 1 apparently do not really match.
L 205 - 207: also the weight per volume of the biomass in each fraction affects the NP synthesis. Next, the resulting NP size and number density (--> spec. surface) affect the antimicrobial activity
L 217: not convinced, different concentrations of different capping agents are compared.
L 228: To my understanding, there is no electrostatic attraction. The cell walls of both Gram-positive and Gram-negative bacteria have an overall negative charge because of the presence of teichoic acids and lipopolysaccharides, respectively, and the capped NPs are negatively charged as well (Zeta potential). Please clarify.
Fig. 6: (How) does the radical formation affect the (stability of the) capping agents in the long term?
L 319: The presence of Ag NPs in the natural environment can pose a threat to entire ecosystem. In this context, Ag-NPs may be toxic to aquatic organisms. Accordingly, I believe that it would be very challenging to get an approval, at least within the EU (https://echa.europa.eu/regulations/nanomaterials-under-bpr). Hence, how could the NPs be removed/recovered when used for dye degradation and wastewater treatment at an industrial scale? Furthermore, in contrast to laboratory conditions, wastewater contains a variety of anions that reduce the activity.
In conclusion, I wonder if apples and oranges are compared here. The extracts differ in the molecular weight range and the total biomass concentration. The molecules differ in reduction potential and affinity. Hence, it is no surprise that these completely different conditions affect the Ag NP synthesis. The average NP size in each fraction has been assessed but in the end the activity is related to the specific surface.
I hope the authors can clarify these issues.
Round 2
Reviewer 2 Report
Accept
Author Response
Reviewer 2 did not comment in this round
Reviewer 3 Report
The authors were anxious to rebut the objections. However I am still not completely convinced.
Figure 3: The histograms (h-l) clearly show that not enough particles have been captured in the TEM for a proper Gaussian distribution. A rough calculation reveales, given the wide size distribution, that at least several thousand particles would have been necessary. That is often not feasible in the TEM but then these values are not really reliable. On the other hand DLS values can be affected by aggregation.
"L 228: To my understanding, there is no electrostatic attraction. The cell walls of both Gram-positive and Gram-negative bacteria have an overall negative charge because of the presence of teichoic
acids and lipopolysaccharides, respectively, and the capped NPs are negatively charged as well (Zeta potential). Please clarify.
Response: The literature suggests that a positive surface charge is an essential condition for AgNPs antimicrobial activity since the positive charge allows for more efficient electrostatic interaction with the negative charges of the bacterial cell wall. However, Maillard et al. (2018) reported that negative charged AgNPs capped with aromatic and hydrophobic moieties synthesized by biological methods could interact via electrostatic attraction with the polar heads of bacteria membrane lipids."
L510-512, "The 510 positively or less negatively charged AgNPs are electrostatically attracted to the negative 511 charged microbial cell wall": But table 1 shows that the AgNP's Zeta potential for all fractions is negative. Apparently, the pH is high (?) - and the Zeta potential is pH dependent. As far as I see, no evidence is presented here that the AgNPs attach to the cell wall. So why, can't the antimicrobial activity simply be mediated by released Ag+ ions? Even the assay in Fig.4 would support this.
